# OBELiX: A Curated Dataset of Crystal Structures and Experimentally Measured Ionic Conductivities for Lithium Solid-State Electrolytes

**Félix Therrien**[1*]    **Jamal Abou Haibeh**[1,2*]    **Divya Sharma**[1]    **Rhiannon Hendley**[3]
**Leah Wairimu Mungai**[1,4]    **Sun Sun**[5]    **Alain Tchagang**[5]    **Jiang Su**[5]    **Samuel Huberman**[2]
**Yoshua Bengio**[1,6]    **Hongyu Guo**[3,5]    **Alex Hernández-García**[1,6]    **Homin Shin**[5]
[1]Mila    [2]McGill University    [3]University of Ottawa    [4]Technical University of Kenya
[5]National Research Council Canada    [6]Université de Montréal
{felix.therrien,alex.hernandez-garcia}@mila.quebec
{hongyu.guo, homin.shin}@nrc-cnrc.gc.ca

## Abstract

Solid-state electrolyte batteries are expected to replace liquid electrolyte lithium-ion batteries in the near future thanks to their higher theoretical energy density and improved safety. However, their adoption is currently hindered by their lower effective ionic conductivity, a quantity that governs charge and discharge rates. Identifying highly ion-conductive materials using conventional theoretical calculations and experimental validation is both time-consuming and resource-intensive. While machine learning holds the promise to expedite this process, relevant ionic conductivity and structural data is scarce. Here, we present OBELiX, a database of ∼600 synthesized solid electrolyte materials and their experimentally measured room temperature ionic conductivities gathered from literature and curated by domain experts. Each material is described by their measured composition, space group and lattice parameters. A full-crystal description in the form of a crystallographic information file (CIF) is provided for ∼320 structures for which atomic positions were available. We discuss various statistics and features of the dataset and provide training and testing splits carefully designed to avoid data leakage. Finally, we benchmark seven existing ML models on the task of predicting ionic conductivity and discuss their performance. The goal of this work is to facilitate the use of machine learning for solid-state electrolyte materials discovery.

## 1    Introduction

Lithium-ion batteries (LIBs) used in most consumer electronics and electric vehicles have seen immense progress in terms of energy density, power density, safety and durability. However, their performance is reaching a plateau. Solid-state batteries are regarded as the next generation of batteries that may allow significant improvement over these characteristics [15, 16]. The key difference between these two technologies is their electrolyte, the medium which allows the transport of ions during charge and discharge. A solid-state electrolyte (SSE)—as opposed to a liquid electrolyte in LIBs—permits new design choices that ultimately lead to better battery properties [3], let alone the fact that they are not flammable unlike their liquid counterparts.

Ionic conductivity ($\sigma$), expressed in siemens per centimeter (S/cm), measures how easily ions can move through a medium or material. Ideal SSEs, also called "superionic" or "fast-ionic" conductors, are electrolytes that exhibit ionic conductivity comparable to those observed in liquid electrolytes and molten solids (> 1 mS/cm). Only a handful of *room temperature* ideal SSEs are known thus

far within a small number of classes of materials: LISICON (e.g., $Li_{14}ZnGe_4O_{16}$), NASICON (e.g., $Li_{1.3}Al_{0.3}Ti_{1.7}(PO_4)_3$), garnet (e.g., $Li_7Li_3Zr_2O_{12}$), perovskite (e.g., $Li_{0.5}La_{0.5}TiO_3$), and argyrodite (e.g., $Li_6PS_5Cl$) [16].

Until now, the discovery of novel SSEs has largely relied on an incremental, experimental approach which consists, for example, of substituting atoms and elements in known compounds. This has allowed the discovery of some highly ion-conductive materials, but greatly limits the search space given that the experimental synthesis and characterization of a new, stable, inorganic solid-state electrolyte is a difficult and costly process that can take months to years [35].

Computational discovery, on the other hand, requires time-consuming atomistic simulations, such as ab initio molecular dynamics (AIMD), to accurately capture the complex relationship between ionic conductivity and the material's structure and composition [6, 22, 4]. These calculations can take from several hours to a few days for a single ionic conductivity and their parameters are often materials specific. Therefore, they are not well suited for large-scale explorations of hypothetical materials.

Machine learning (ML) has the potential to greatly accelerate the discovery of novel SSEs. Naturally, it can be used to predict ionic conductivity directly using, for example, graph neural networks (GNNs), which have been used extensively and successfully in materials science [24, 5]. Machine-learned force fields or interatomic potentials (MLFF or MLIP) can also be used to obtain ionic conductivity through molecular dynamics in the "classical" way while using significantly less resources [31]. Finally, generative frameworks can accelerate dynamics simulations [21] and, provided that good ionic conductivity models are developed, there exists a wide range of frameworks that could generate new materials conditioned on that property [12, 36, 34, 20]. However, the main obstacle to the development and validation of these models—and to some extent theoretical models—is the scarcity of relevant experimental ionic conductivity and structural datasets. Indeed, as detailed in the next section, the few datasets that exist contain partial material descriptions and ionic conductivity measurements at various or unspecified temperatures. To the best of our knowledge there does not exist another open access dataset of experimental room temperature ionic conductivities with corresponding full crystal descriptions.

In this work, we assembled OBELiX (Open solid Battery Electrolytes with Li: an eXperimental dataset), a curated database of 599 synthesized solid electrolyte materials and their experimentally measured room temperature ionic conductivity along with descriptors of their space group, lattice parameters, and chemical composition[1]. The database is analyzed in terms of the distribution of ionic conductivity, space groups, elements, and repeated compositions. We also propose a training and testing split that avoids data leakage between similar entries while balancing distributions of properties across splits. We use this split to benchmark the performance of 7 machine learning models at directly predicting room temperature ionic conductivity ($\sigma_{RT}$).

We believe that this dataset and benchmark can significantly spur the use of ML for the discovery of novel solid-state battery materials. The size seems small but it is important to realize that the database represents a large fraction of all materials whose ionic conductivity has been characterized experimentally. Importantly, this database has been carefully curated by domain experts and formatted by machine learning scientists to facilitate its use by this community. Finally, we believe that this benchmark can encourage novel machine learning research tailored to low-data regimes.

## 2   Related work

Crystal structure databases such as the Materials Project [13] or the Inorganic Crystal Structure Database (ICSD) [2, 11] contain large amounts of potential candidates for solid-state electrolytes. For example, Sendek et al. [28] screened more than 12,000 Li-containing crystals for Li-ion SSEs using multiple criteria, thereby identifying 317 candidates, among which 21 crystals that showed promise as SSEs were selected from an ML-guided model. The ionic conductivity of these 21 structures was estimated theoretically. Jalem et al. [14] annotated 318 compounds by calculating ion migration energy barriers ($E_b$), a less accurate but computationally lighter property that relates to ionic conductivity. Bayesian optimization was employed to screen candidate compounds with low $E_b$. He et al. [10] compiled a database of over 90,000 crystal structures, including more than 7,000 structures with preliminary ion-transport data obtained through geometric analysis, and 12,000

---

[1]OBELiX is available here: github.com/NRC-Mila/OBELiX

Table 1: Comparison of our dataset (OBELiX) with existing ones based on key features and labels. For features, the numbers represent the number of entries with that feature that are labeled with at least one experimental or computational ion transport property (not necessarily ionic conductivity). Numbers in parentheses represent proprietary or private data.

| Dataset | Labels | | Features | | | |
| | $\sigma_{RT}^{\exp}$ | $\subset \sigma^{\exp}$ | Comp. | Spg | Lattice | CIFs |
| --- | --- | --- | --- | --- | --- | --- |
| Sendek et al. | 0 | 0 | 317 | 317 | 317 | 317 |
| Jalem et al. | 0 | 0 | 318 | 318 | 318 | 318 |
| He et al. (SPSE) | 0 | 0 | 75 (12k) | 75 (12k) | 75 (12k) | 75 (12k) |
| Hargreaves et al.(LiIon) | 465 | 820 | 820 | 0 | 0 | 0 |
| Laskowski et al. | 1346 | 1346 | 1346 | 0 (344) | 0 (344) | 0 (344) |
| Shon and Min | n.a. | 4032 | 4032 | 0 | 0 | 0 |
| Yang et al. (DDSE) | (1939) | (2448) | 2448 | 0 | 0 | 0 |
| OBELiX | 599 | 599 | 599 | 599 | 599 | 321 |

activation energy values ($E_b$) calculated using the bond valence site energy method. Additionally, they manually extracted 75 CIF files from literature data. They employed empirical and geometrical methods to estimate the minimum energy paths of these structures and obtain $E_b$, but they did not predict $\sigma$.

On the exprimental side, the Liverpool Ionics (LiIon) Dataset [9] reports 820 entries containing chemical composition, structural family, and ionic conductivity at different temperatures (from 5 to $873°C$) measured by alternating current impedance spectroscopy, among which 465 entries were at room temperature. Laskowski et al. [17] gathered a dataset of 1346 entries with compositions, space group, and corresponding $\sigma_{RT}$, with a subset of 344 compounds whose structures are manually matched with an ICSD ID. The full dataset, including references, is only available as a pdf file. Shon and Min [29] used text mining to extract more than 4000 ionic conductivity measurements from 1457 papers. Each ionic conductivity measurement is associated with a composition and about 350 are also associated with a "structure type". Measurement temperature is not specified and compositions are not always fully described. A recent study by Yang et al. [33] introduced the Dynamic Database of Solid-State Electrolyte (DDSE) to facilitate the exploration of structure-performance relationships and accelerate the discovery of high-performance solid-state electrolytes (SSEs). The database contains performance data for 2448 materials (at time of writing), including ionic conductivity obtained from experimental reports, across a broad temperature range (132.40–1261.60 K). Ionic conductivity data is only available upon request to the authors.

These recent reports greatly increased the amount of readily available experimental ionic conductivity data. However, they contain limited structural information: the databases by Shon and Min [29] and Yang et al. [33] contain only a qualitative structure description for some materials, the LiIon dataset only includes the structural family and the dataset by Laskowski et al. [17] is limited to space group information. Although the full crystallographic information of the 344 compounds of the Laskowski dataset for which the ICSD ID is provided could be retrieved, the proprietary ICSD is not available to most researchers in the ML community. Table 1 summarizes the differences in terms of available features across the databases discussed above.

The lack of precise structural information labeled with ionic conductivity makes it difficult (1) to compare experimental values with theoretical predictions which require full crystal descriptions and (2) to train machine learning models to accurately predict ionic conductivity.

## 3 Data

### 3.1 Background

All the solid-state electrolyte materials in OBELiX are crystal structures. Crystals are materials with a repeating arrangement of the same atoms. The composition (or chemical formula) describes which atoms are present in what proportion. The repeating pattern in a crystal, the unit cell, is contained within a parallelepiped (lattice) with edges $a$, $b$, $c$ and angles $\alpha$, $\beta$, $\gamma$ that, together, form

the lattice parameters. The symmetry of a crystal is described by its space group, of which there are 230, representing all possible combinations of symmetry operation in 3D. Space groups are usually denoted as a string of numbers and letters representing their symmetry operations or a number between 1 and 230, e.g. $Fm\bar{3}m$ for space group 225.

While the combination of the composition, lattice parameters and space group is often sufficient to qualify materials, they do not fully describe the crystal structure because in general they do not specify the positions of each atom. Some experimental papers perform an analysis (Rietveld refinement) of the X-ray powder diffraction pattern of their materials to estimate atomic positions. This information is necessary to perform molecular dynamics simulations, for example.

In contrast to theory-based data found in the Materials Project, for example, experimental compositions often feature fractional numbers (real numbers rather than integers) resulting from partially vacant sites or disorder associated with partial cation substitution. Consider, for example, composition $K_{0.1}Li_{0.9}SbO_3$. At a specific location in the crystal (a site) there is a 90% probability of finding a lithium (Li) atom and a 10% probability of finding a potassium (K) atom. Site occupancy does not need to add up to one since sites are often partially empty.

Such partial occupancy is ubiquitously observed in Li-ion SSEs [19] and it plays a crucial role in creating diffusion pathways. For example, the $\sigma_{RT}$ of tetragonal $Li_7Li_3Zr_2O_{12}$ with a space group of I41/acd (no. 142) is two orders of magnitude smaller than that of the same garnet framework of cubic $Li_7Li_3Zr_2O_{12}$ with Ia-3d (no. 230) (see Figure 1a). In this case, the disordering and partial occupation of Li (at the 96h site) promotes the Li-ion conduction. In the halide structure of $Li_3InCl_6$ (Figure 1b), the substitution of one Li+ with the In3+ cation introduces two intrinsic vacancies, to which is attributed the high $\sigma_{RT}$ of that material. In sum, in order to screen SSEs with high $\sigma_{RT}$, it is highly desirable to include partial occupancy as a key feature of the materials.

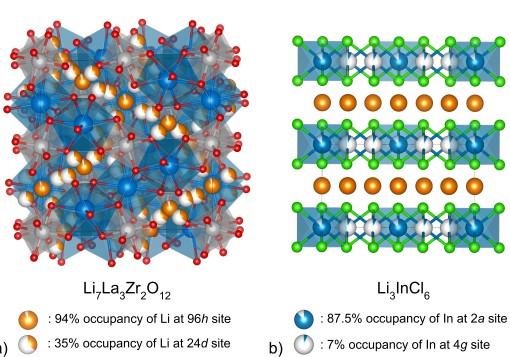

Figure 1: Examples of solid state electrolyte materials with partial occupancies.

## 3.2 Data collection

We built our dataset starting from the Liverpool Ionics Dataset and the Laskowski dataset by selecting materials for which the experimental room temperature ionic conductivity, space group and lattice parameters could be obtained. We manually retrieved missing information (e.g. lattice parameters or $\sigma_{RT}$) from the original paper's table or figures. Through this procedure, we obtained a total of 599 distinct entries including an additional 15 entries from other sources. Figure 2b shows the number of common entries between these two datasets and ours.

Ionic conductivity is usually reported as a property of the materials in the powder form, which includes the effect of defects and grain boundaries. It is referred to as "total" ionic conductivity. The ionic conductivity of individual grain is sometimes reported as the "bulk" ionic conductivity. When both were available we recorded both. This is relevant because the total ionic conductivity of materials not only depends on their crystal structure but also on factors such as the size of particles.

For each material, we recorded the total composition including the number of formula unit Z. For example, the unit cell compositions of $Li_3PO_4$ could be $Li_6P_2O_8$ and $Li_{12}P_4O_{16}$ with Z=2 for the space group pnm21 (no. 31) and Z=4 for pnma (no. 62), respectively. This added information makes the computation of density and volumetric density possible for every material in the dataset.

To the best of our capacity, we have ensured that the reported structural information in OBELiX corresponds exactly to the same material for which the ionic conductivity was measured. We also filtered the dataset for exact duplicates and ensured that near duplicates were truly different materials. It is common for papers to report ionic conductivity measured elsewhere when synthesizing a material and vice versa for structural information. If not caught, this can lead to two entries with the exact same ionic conductivity, only one of which is the actual material for which it was measured.

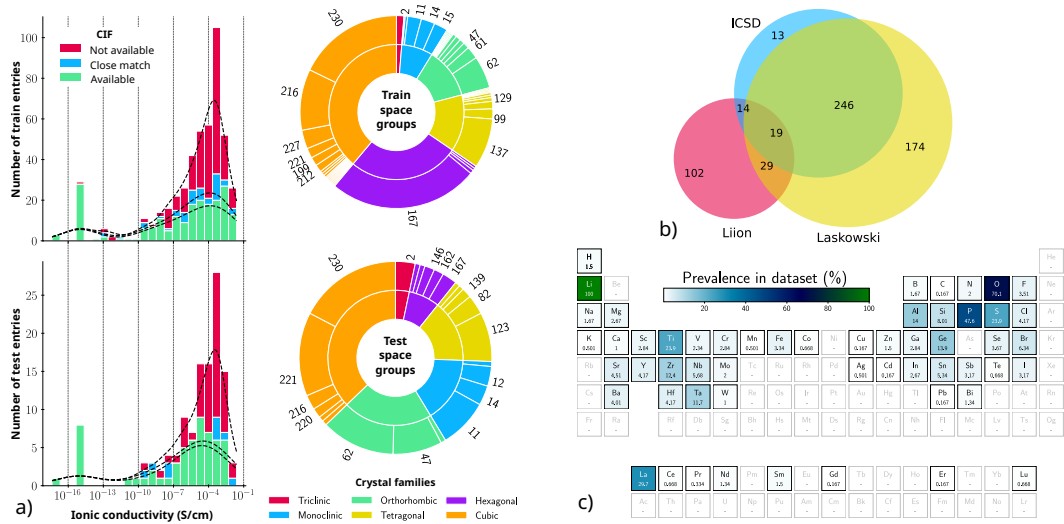

Figure 2: a) Distributions of ionic-conductivity values for the training and testing sets along with proportions of crystal families and space groups. Only space groups that represent more than 1% of the sets are labeled. b) Venn diagram showing shared entries between our dataset and others c) Proportion of entries that contain each element in the periodic table. Elements that are not present in the dataset are shaded. Generated with pymatviz [23].

The ICSD is a large database of experimental data in the form of crystal information files (CIF) that contain full crystal descriptions including atomic positions. Given that a significant portion of publications in this field have crystal information in the ICSD, we searched the database for all entries matching the lattice, parameters composition and associated publication. We found 234 exact matches with our entries, for which we obtained the CIFs. We also manually retrieved crystal information for 27 entries. Finally, we searched the ICSD and the Materials Project for structures that matched the space group and closely matched the composition ($\pm$ 0.05) and lattice parameters ($\pm$ 3%) of our entries and found 60 additional CIF files (labeled as close matches). This forms a total of 321 entries with CIF information.

Because the ICSD is a proprietary database, we are not able to publish 292 of the CIF files and can only link our entries to their corresponding ICSD ID. However, to reach a broader audience, in agreement with the ICSD, we openly publish a set of 292 CIF files for which a normally distributed random noise with standard deviation 0.01 ($\epsilon \sim N(0, 0.01)$) in fractional coordinates was added to the original atomic positions. This noise was added while making sure that the full symmetry of the crystal was preserved. We measured the effect of noise on model performance (see section 4) and found that it made little to no difference (see the SI for more details).

### 3.3 Data splits

Experimental papers in this field often measure ionic conductivity for several variations of the same materials while changing the composition slightly. This can lead to multiple entries that are very similar and often have similar ionic conductivities. There are also several entries in our dataset that have the same composition, which may also lead to similar ionic conductivities. To avoid data leakage when testing machine learning models on OBELiX and to fairly compare new models in the future, we provide a split of the data where entries from the same paper or that have the same composition must be in the same set (training or testing).

To obtain this split, we used a Monte Carlo method that moved groups of entries from one set to the other to minimize (1) the difference between the distribution of log ionic conductivity between the two sets and (2) the difference between their respective subsets containing CIF files. The algorithm also ensured that the final test set represented between 20% and 30% of the data. The obtained distribution of log ionic conductivity in each set and subset is presented in Figure 2a along with the proportion of each crystal family and space group. The test set represents 20.2% of the full dataset and 20.9% of the subset that has CIF files.

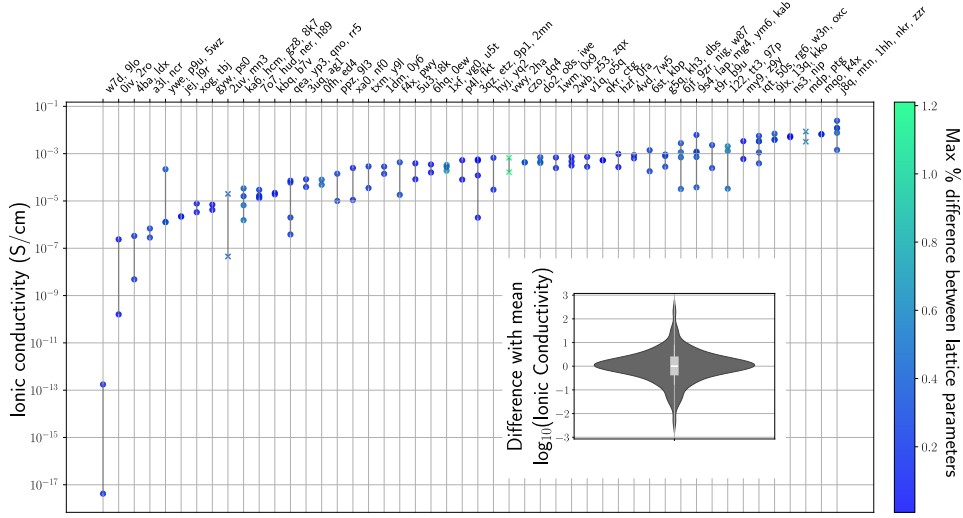

Figure 3: Ionic conductivity of entries in the dataset that have the same composition and space group. The color shows the largest relative difference between lattice parameters within a set of entries with same space group and composition. The inset shows the distribution of differences with the mean ionic conductivity of the sets in log scale. It is scaled proportionally to the rest of the plot.

The distributions in log space of ionic conductivity for the two sets are very similar. Note that the entries plotted at $10^{-15}$ were reported as having a conductivity of "less than $10^{-10}$" without a quantitative value. The proportion of crystal families and space groups is also fairly similar between the two sets, except for space group 167, which is much more prevalent in the training set. This is due to the fact that a large group of entries (106) with space group 167 were either from the same paper or had the same composition. This meant that the entire group could not be split between the two sets without leaking either a paper or a composition.

The dataset contains 55 space groups, 4 of which are only in the test set. Figure 2c shows the prevalence of the 55 different elements that are present in the dataset. All entries contain lithium (by design) and most of them contain oxygen. Phosphorus, lanthanum, sulfur and titanium follow as the most prevalent elements. Silver is the only element that is not found in the training set (it is only in the test set).

About 75% (245/321) of the entries with atomic information have some level of partial occupancy (disorder). The proportion of partially occupied structures in each split was not controlled for explicitly, but it is similar in the test (53/67) and train (192/254) splits. For the rest of the entries, when atomic positions and occupations are unknown, it is not always possible to tell if a structure is disordered.

## 4 Benchmarks

In this section, we benchmarked how well existing models perform on the new dataset. This evaluation is essential for determining whether these models can be effectively applied or if there is a need to develop new models better suited for the task.

We note that experimental data intrinsically embeds errors and uncertainty associated not only with various sources of measurement techniques but also with data extraction from figures and inconsistent labeling (e.g., bulk, grain boundary, or total ionic conductivity are often indistinguishably reported). Before assessing the performance of predictive models it makes sense to quantify the uncertainty ("performance") of experimental data acquisition. Thankfully, our dataset contains 48 sets of compositions and space groups that have multiple entries, spanning a total of 122 entries. These entries and their corresponding ionic conductivities are plotted in Figure 3. The color represents the maximum difference in lattice parameters between any two entries of a same set. The maximum difference is of only 1.2% for all sets, which gives us confidence that grouped materials are in fact the same. This means that these materials were synthesized and their ionic conductivity measured two or

more times, most likely by different researchers. This represents a unique opportunity to quantify experimental uncertainty and reproducibility. The inset of Figure 3 shows the distribution of log ionic conductivities with respect to the mean of each set of repeated materials. The root mean squared deviation from the set averages is of 0.63 log(S/cm) and the mean absolute deviation from the set medians is of 0.41 log(S/cm). The latter can be compared to the model's mean absolute error when predicting log ionic conductivity and represents its lower bound. Therefore, any model that would be reported as having lower MAE that that value would most likely be over-trained.

## 4.1 Baselines

To evaluate the performance of ML models on OBELiX, we tested five widely adopted graph neural networks developed specifically for materials science applications, PaiNN [25], SchNet [27], M3GNet [7], SO3Net [26], and CGCNN [32] on the subset of the dataset that contains CIF files. These graph-based models, where each node represents an atom, effectively capture atomic interactions while preserving molecular invariance, enabling accurate material property predictions when trained on large datasets [18]. On the full dataset, where atomic positions are not always available, we also tested two standard machine learning models, a random forest (RF) and a multilayer perceptron (MLP).

The RF and the MLP use the composition, space group and lattice parameters as inputs where the composition is a vector containing the occurrence ($\in \mathbb{R}$) of each element of the periodic table. The 3D geometric models use the crystal structure as their input and build different representations from that structure. The crystal structure contains the composition and space group information implicitly, but the models are not given that information explicitly. None of the model can take into account partial occupancy of the sites, therefore occupations are rounded to the nearest integer before being fed to the models.

## 4.2 Setup

To optimize the training process and assess the stability of the models, we implemented a 5-fold cross-validation strategy. For hyperparameter optimization, we employed a grid search strategy across a predefined space of 100 randomly sampled hyperparameter sets for each model. This number was selected to strike a balance between comprehensive exploration of the hyperparameter space and computational feasibility. In the case of RF and MLP where training is extremely fast; all hyperparmeter sets were tested. The hyperparameter space was carefully designed for each model based on its unique architecture and requirements (see Table S2 in the SI for a complete list). For example, PaiNN's search space included parameters such as the cutoff distance, number of interactions, and batch size.

We computed the mean absolute error (MAE) between the predicted and the measured ionic conductivities to evaluate the performance of each configuration. Specifically, the average validation MAE across all folds in the cross-validation process was used to assess each setup's effectiveness. The hyperparameter set that achieved the lowest average validation MAE was selected as the best-performing configuration. After choosing the best hyperparameters, each model was retrained on the entire training set and evaluated on the test set. A detailed table of the selected hyperparameters for each model is included in the SI (Table S2).

Pretraining can enhance model performance by initializing weights with knowledge from larger datasets and related tasks, which is then fine-tuned on a smaller, task-specific dataset. We pretrained PaiNN and SchNet on the Materials Project with a band gap prediction task. In this case we fixed the trained representation (PaiNN or SchNet) and trained the output model (an MLP followed by a pooling layer) on OBELiX. For M3GNet and CGCNN we use pretrained models that were available on their public repositories. The M3GNet model was trained on formation energy per atom whereas CGCNN was trained on Fermi energy both from the Materials Project. As recommended in their respective documentation, we fine-tuned the models by training all model parameters starting from the trained models.

## 4.3 Discussion

Figure 4 and Table S1 present our benchmarking results and Figure S1 presents the corresponding parity plots. The MLP and the RF were trained on the full training set, but tested on both the full

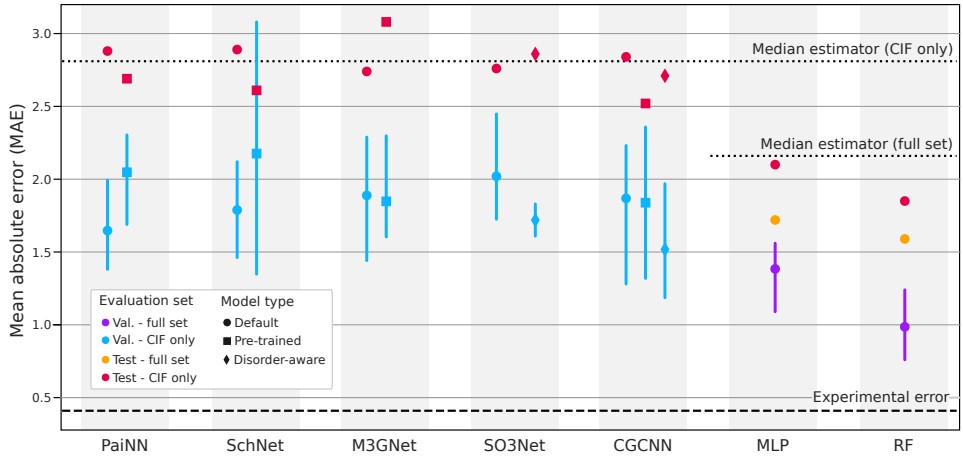

Figure 4: Benchmarking of various ML models. The same data is tabulated in Table S1. Simpler models outperform geometric GNNs.

test set (in orange) and the subset of the test set that has CIF files (in red). The goal is to be able to compare their performance directly with geometric models, given that the variance of the CIF subset is larger.

The two simple models, RF and MLP, outperform all 3D geometric models both in the cross-validation and the test performance even when comparing with the subset of the test set that has CIF files. There are two factors that could explain this result. First, the RF and the MLP used the full training set of 478 structures while the other models were limited to the subset of 254 entries that have CIF files. Second, the geometric models use crystal information to infer properties of the crystal, but they do not properly handle partial occupancies which, as discussed before, are very common in SSE materials and are present in about 3/4 of our CIF files. In order to use these models without modification on our dataset we rounded occupancies to the nearest integers which can lead to important changes in the composition.

To partly verify the above claim that dataset size and the presence of partial occupancy can explain the increased performance of the simple models, we retrained them on the subset of entries that have CIF files only. Doing so, the MAE of the MLP increased to 3.15 while that of the RF was maintained at 1.87. Therefore, dataset size does seem to have a significant impact on the MLP and may explain the difference in performance between that model and the larger models. Random forest still performs well even given less data. Rounding compositions to the nearest integer on the other hand, had little effect on both the RF and the MLP. Rounding compositions is *similar* to rounding site occupancy, but it does not have exactly the same effect. Nevertheless, it indicates that the absence of partial occupancy likely does not explain the difference in performance between the simple models and the more complex ones.

To further explore the effects of partial occupancy, which, as explained in Section 3.1, is an important concept in this field, we introduce new implementations of both CGCNN and SO3Net (dis-CGCNN and dis-SO3Net) that take into account partial occupation (disorder). In both cases, the atomic embedding is replaced with a *site* embedding that is an average over the element embeddings weighted by occupancy. We trained these models using the same optimal hyperparameters as their original version. The results presented in Figure 4 and at the bottom of Table S1 show a small improvement in cross-validation performance but it does not translate into significantly better test performance.

The 3D geometric models not only performed poorly compared to simple ML models using less structural information, but their performance on the test set was barely better or sometimes worse than predicting the median of the training set (doted line in Figure 4. This shows that these large models can easily overfit small experimental datasets which was also observed in other studies [8]. Moreover, given that the cross-validation splits were chosen randomly within the training set and that the test set was build using the method described in Section 3.3, the relatively large difference in performance between the validation and testing sets illustrate the importance of carefully building

leakage-free test sets and that choosing the test set randomly would have most likely led to a false impression of performance.

Finally, pretraining of 3D geometric models offers some marginal improvements for PaiNN, SchNet and CGCNN. As mentioned in Section 4.2, the pretraining of PaiNN and SchNet restricts the trainable model size which may reduce accuracy while increasing generalizability. This would explain their slightly higher validation MAE and lower training MAE. A better pretrained representation might compensate for the reduced expressivity and increase both accuracy and generalizability, but a much more in-depth analysis of the possible pretraining labels and datasets would be required. In the case of CGCNN and M3GNet which were fine-tuned by allowing all model parameters to change, it is possible that the pretraining property used for CGCNN was "closer" (or more relevant) to ionic conductivity which allowed it to stay in the same weight "basin" and take advantage of the pretrained model's generalizability. It is important to bear in mind that the variability of the prediction accuracy is high in this small data regime as illustrated by the validation MAEs' standard deviations and that much of the difference between these models falls within that variability. Therefore one must be careful when interpreting the results.

## 5    Limitations

We have built OBELiX as carefully as possible making sure that all features match the measured ionic conductivity correctly. However, since data is reported and measured in very different ways across journals and decades, there most probably remains inconsistencies between some of the entries especially in terms of atomic positions which are particularly difficult to measure and report. We will continue to improve the dataset as these issues come to light.

OBELiX is small for ML standards. The difficulty of building an experimental dataset is that there is only a limited number of experiments that were actually performed. Section 4 shows how challenging it is to train existing models on such a small data regimes. Ultimately, it highlights the need for models, training architectures and benchmarks tailored for small data regimes, that could benefit numerous applied fields with similarly limited experimental data (e.g. [1]).

We benchmarked ionic conductivity prediction on our dataset with popular existing models *as is* and using standard training and hyperparameter tuning. We are aware that performance could be improved by modifying the model architectures, training procedure or with data augmentation, but we consider that these methods would not be "baselines" and are outside the scope of this paper.

## 6    Conclusion and outlook

In this paper, we presented OBELiX, a dataset of 599 materials with experimental room temperature ionic conductivities curated by domain experts, including 321 structures with full crystallographic information. We gathered these materials from existing databases and manually extracted data from the literature to build a consistent, easy-to-access database of solid-state electrolyte materials. We benchmarked several ML models and found that the simple random forest model had the best predictive performance. Modern geometric GNNs on the other hand, likely over-fit and were unable to perform well on our carefully designed test set. These findings highlight the immense opportunity for improvement in ML methods specific to this task and tailored for low data regimes.

We hope that OBELiX will serve as a reference point to train and test ionic conductivity models for the ML and computational materials science community in general, ultimately advancing solid-state battery technology.

## 7    Data availability

All data is freely available on our public repository[2] as a single csv or xlsx file accompanied by a set of 321 CIF files, including 291 with added random noise. The same data is also available on Kaggle[3]. All experiments were performed with OBELiX version 1.0.0 [30].

---

[2]github.com/NRC-Mila/OBELiX

[3]www.kaggle.com/datasets/flixtherrien/obelix

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

# Supplemental Information

## A  Parity plots and data from Figure 4

Table S1 contains the information from Figure 4 in a tabulated form. Figure S1 presents parity plots for benchmarking experiments discussed in Section 4.

## B  Baseline models

We trained and tested 7 ML models which are briefly described below:

1. RF, as an ensemble of decision trees, is robust to noisy data and provides feature importance insights, making it a strong baseline for structured datasets.

2. MLP, a neural network-based approach, captures complex nonlinear relationships, offering a comparison to deep learning-based methods.

3. PaiNN [25] enforces E(3)-equivariance, enabling accurate modeling of atomic interactions and force predictions.

4. SchNet [27] learns continuous filter representations, making it effective for capturing atomic environments.

5. M3GNet [7]integrates message passing with three-body interactions, improving property predictions for crystalline materials.

6. SO3Net [26] leverages spherical harmonics to enhance equivariant representations for molecular and solid-state systems.

7. CGCNN [32] models crystal structures directly as graphs, making it a strong baseline for learning structure-property relationships.

Table S2 shows the best hyperparameter sets for each model presented in Table S1

## C  Effects of added random noise

Table S3 presents the cross-validation and test MAEs for the models trained on randomized atomic positions. The models were trained on the randomized data and tested on the original CIFs. There is no significant difference in performance between models trained on the original data and models trained on data with added random noise on atomic positions.

## D  Computational resources used for benchmarking

Table S4 presents the resources used to find optimal hyperparameters and train each model.

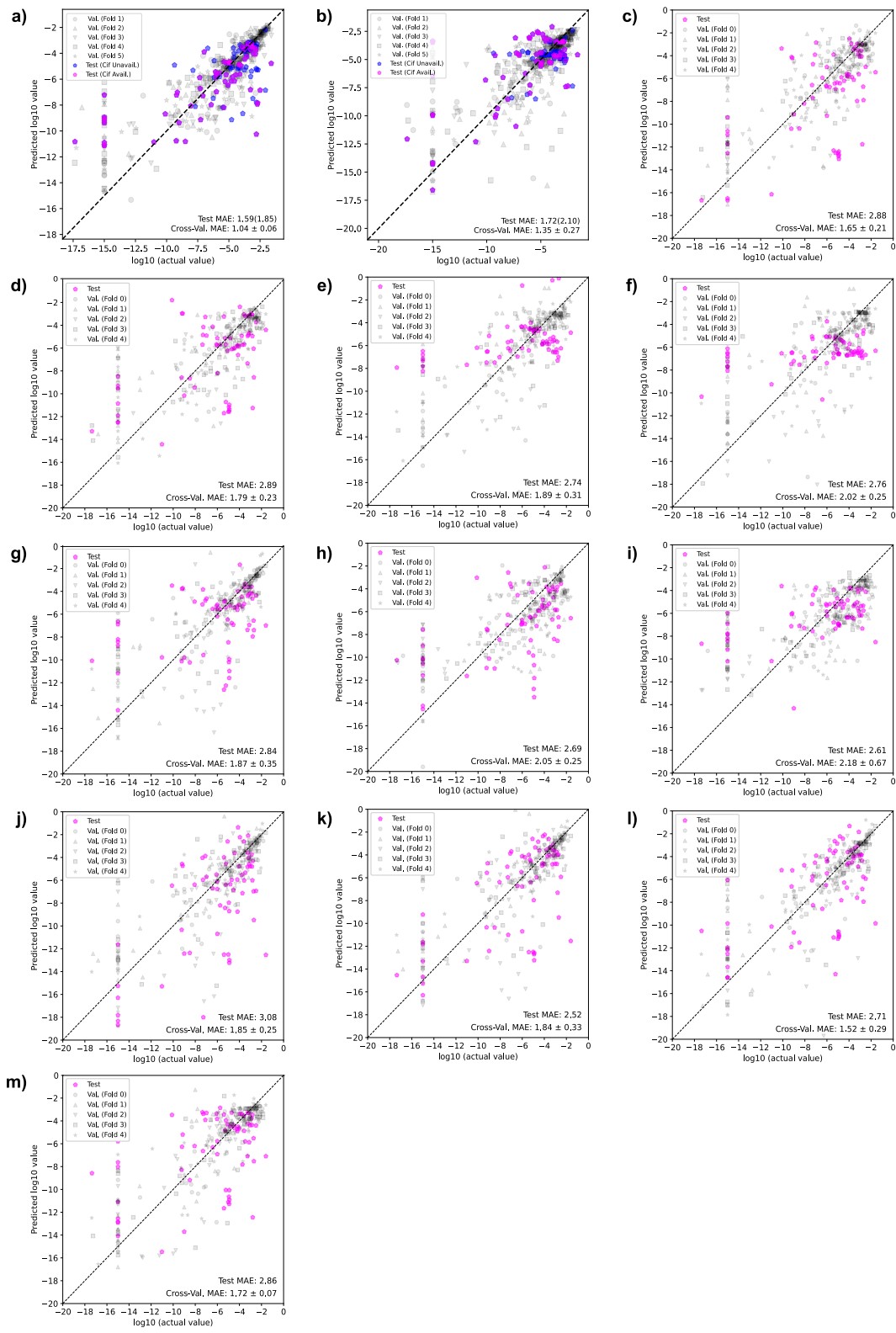

Figure S1: Parity plots for benchmarked models. a) Random Forest b) Multilayer perceptron c) PaiNN d) SchNet e) M3GNet f) SO3Net g) CGCNN h) PaiNN with pretraining i) SchNet with pretraining j) M3GNet with pretraining k) CGCNN with pretraining l) CGCNN with disorder (partial occupancy) m) SO3Net with disorder (partial occupancy)

Table S1: Benchmarking of various ML models with and without pretraining. For the median prediction, the random forest (RF) and the multilayer perceptron (MLP), results are presented for the full dataset and numbers in parenthesis are results for the subset of the test set that has CIF files. All other results apply only to entries with CIF files. "p-" indicates a model that was pretrained and "dis-" indicates a model that was modified to take partial occupancy (disorder) into account.

| Model | Cross-val. MAE Avg. $\pm$ SD | Test MAE |
|---|---|---|
| Experiment | | 0.41 |
| Median pred. | | 2.16 (2.81) |
| RF | $1.04 \pm 0.06$ | 1.59 (1.85) |
| MLP | $1.35 \pm 0.27$ | 1.72 (2.10) |
| PaiNN | $1.65 \pm 0.21$ | 2.88 |
| SchNet | $1.79 \pm 0.23$ | 2.89 |
| M3GNet | $1.89 \pm 0.31$ | 2.74 |
| SO3Net | $2.02 \pm 0.25$ | 2.76 |
| CGCNN | $1.87 \pm 0.35$ | 2.84 |
| p-PaiNN | $2.05 \pm 0.25$ | 2.69 |
| p-SchNet | $2.17 \pm 0.68$ | 2.61 |
| p-M3GNet | $1.85 \pm 0.25$ | 3.08 |
| p-CGCNN | $1.84 \pm 0.33$ | 2.52 |
| dis-CGCNN | $1.52 \pm 0.29$ | 2.71 |
| dis-SO3Net | $1.72 \pm 0.07$ | 2.86 |

Table S2: The selected hyperparameters for the baseline models.

| Model | Hyperparameter | Value | Model | Hyperparameter | Value |
|-------|----------------|-------|-------|----------------|-------|
| RF | `max_depth` | 36 | PaiNN | `cutoff` | 5 |
| | `max_features` | sqrt | | `n_interactions` | 2 |
| | `min_samples_leaf` | 1 | | `n_atom_basis` | 80 |
| | `n_estimators` | 50 | | `batch_size` | 32 |
| MLP | `activation` | relu | | `max_epochs` | 100 |
| | `batch_size` | 16 | | `weight_decay` | 0.0001 |
| | `early_stopping` | True | Schnet | `cutoff` | 5 |
| | `hidden_layer_sizes` | [64, 64, 64, 64] | | `n_interactions` | 3 |
| | | | | `n_atom_basis` | 80 |
| | `learning_rate` | adaptive | | `batch_size` | 32 |
| | `learning_rate_init` | 0.01 | | `max_epochs` | 100 |
| | `max_iter` | 1000 | | `weight_decay` | 0.01 |
| | `n_iter_no_change` | 100 | M3GNet | `cutoff` | 5.0 |
| | `solver` | adam | | `threebody_cutoff` | 5.0 |
| | | | | `is_intensive` | True |
| | | | | `readout_type` | "set2set" |
| | | | | `nblocks` | 3 |
| | | | | `dim_node_embedding` | 128 |
| | | | | `dim_edge_embedding` | 128 |
| | | | | `units` | 64 |
| | | | | `batch_size` | 35 |
| | | | | `max_epochs` | 50 |
| | | | | `lr` | 0.001 |
| | | | | `weight_decay` | 0.01 |
| | | | SO3Net | `cutoff` | 5.0 |
| | | | | `is_intensive` | True |
| | | | | `nmax` | 2 |
| | | | | `lmax` | 1 |
| | | | | `target_property` | "graph" |
| | | | | `readout_type` | "set2set" |
| | | | | `nblocks` | 3 |
| | | | | `dim_node_embedding` | 64 |
| | | | | `nlayers_readout` | 3 |
| | | | | `units` | 32 |
| | | | | `batch_size` | 35 |
| | | | | `max_epochs` | 80 |
| | | | | `lr` | 0.001 |
| | | | | `weight_decay` | 0 |
| | | | CGCNN | `n_conv` | 3 |
| | | | | `n_h` | 1 |
| | | | | `atom_fea_len` | 64 |
| | | | | `h_fea_len` | 64 |
| | | | | `batch_size` | 35 |
| | | | | `epochs` | 50 |
| | | | | `lr` | 0.001 |
| | | | | `weight_decay` | 0 |

Table S3: Performance of the 5 geometric models on the public dataset with added random noise to the atomic positions. Test results are on the original test set.

| Model | Cross-validation MAE | Test MAE |
|--------|:---:|:---:|
| PaiNN | $2.03 \pm 0.27$ | 2.95 |
| SchNet | $1.99 \pm 0.22$ | 2.78 |
| M3GNet | $1.83 \pm 0.29$ | 2.91 |
| SO3Net | $1.98 \pm 0.23$ | 2.79 |
| CGCNN | $1.94 \pm 0.42$ | 2.95 |

Table S4: Resource usage for benchmarking

| Model | Hardware | Hyperparameter tuning | Final training |
|:---:|:---:|:---:|:---:|
| RF | AMD EPYC 7502 (1 core) | 7min | 1s |
| MLP | AMD EPYC 7502 (1 core) | 50min | 14s |
| PaiNN | NVidia A100 GPU | 2h40min | 2min |
| SchNet | NVidia A100 GPU | 1h55min | 2min |
| M3GNet | NVidia A100 GPU | 3h35min | 2min |
| SO3Net | NVidia A100 GPU | 2h25min | 2min |
| CGCNN | NVidia A100 GPU | 1h40min | 1min |

