# OpenReview forum: "OBELiX: A Curated Dataset of Crystal Structures and Experimentally Measured Ionic Conductivities for Lithium Solid-State Electrolytes"
_NeurIPS.cc/2025/Datasets_and_Benchmarks_Track — Submitted to NeurIPS 2025 Datasets and Benchmarks Track_

### Official Review · Reviewer_hHZL · 2025-06-16

**Ethics Flags:** Safety and security, Environmental im…
**Rating:** 5
**Confidence:** 4

**Summary:**

The authors note that identifying highly ion-conductive materials using conventional theoretical calculations and experimental validation is both time-consuming and resource-intensive. To this end, the authors consider leveraging the power of machine learning for this problem, and they propose a novel database of 600 synthesized solid electrolyte materials and their experimentally measured room temperature ionic conductivities.

**Additional Feedback:**

Please see the Limitations Weaknesses textbox.

**Dataset Code Accessibility:**

Yes

**Dataset Code Comments:**

The authors have provided the dataset and the related source codes.

**Ethical Considerations:**

No, there are no or only very minor ethics concerns

**Final Justification:**

I have read the rebuttal, and I will maintain a positive rating.

**Limitations Weaknesses:**

- On Page 2, the authors write: "for example, graph neural networks (GNNs), which have been used extensively and successfully in materials science." Here, some related typical GNN architectures can be cited, such as the literate *Graph substructure assembling network with soft sequence and context attention." IEEE Transactions on Knowledge and Data Engineering 35.5 (2022): 4894-4907.* can discover the key component in molecular graphs.


- On Page 4, the last sentence seems to be incomplete: "This information is necessary to perform molecular dynamics simulations, for example.".

- The text at the top of Figure 3 is hard to read.

**Strengths Contributions:**

- The proposed dataset is novel and can facilitate the application of machine learning in materials science.

- The dataset and the source codes of the benchmark are provided, which is very convenient for other researchers to try out, promoting research in this field.

- The experimental evaluation is comprehensive.

- The presentation is good.

---

> ### Author Rebuttal · Authors · 2025-07-31
>
> Thank you for your positive review and feedback. We appreciate that you think our dataset can "facilitate the application of machine learning in materials science.'' and that the availability of our data will promote research in this field.
>
> ### Comment 1
> > On Page 2, the authors write: "for example, graph neural networks (GNNs), which have been used extensively and successfully in materials science." Here, some related typical GNN architectures can be cited, such as the literate Graph substructure assembling network with soft sequence and context attention." IEEE Transactions on Knowledge and Data Engineering 35.5 (2022): 4894-4907. can discover the key component in molecular graphs.
>
> Thank you for pointing out this work, we will add it as a reference in the final version.
>
> ### Comment 2
> > On Page 4, the last sentence seems to be incomplete: "This information is necessary to perform molecular dynamics simulations, for example.".
>
> The sentence was indicating that "molecular dynamics simulations" was an example of something that requires "This information". It was not clear and we will rephrase it as such:
>
> >> The full-crystal description including atomic positions is the information required to perform, for example, molecular dynamics simulations or density functional theory calculations.
>
> ### Comment 3
> > The text at the top of Figure 3 is hard to read.
>
> We will increase the overall size of the figure and the size of the entry labels in the final version.

---

### Official Review · Reviewer_set9 · 2025-06-22

**Rating:** 5
**Confidence:** 5

**Summary:**

This paper introduces OBELiX, a curated dataset of 599 experimentally measured lithium-based solid-state electrolytes, including room-temperature ionic conductivity and crystal structure (CIF) data for 321 entries. The dataset includes consistent composition, symmetry, and lattice parameters, and is split into train/test sets with careful control to avoid data leakage. The authors benchmark seven machine learning models including graph-based and conventional approaches for direct prediction of ionic conductivity.

**Additional Feedback:**

While the paper is already well-structured and likely to be accepted, I provide the following additional feedback to further enhance its completeness.

1. **Missing recent GNN baselines**: While the benchmark includes representative models like SchNet, PaiNN, and M3GNet, it omits more recent architectures such as NequIP, MACE, GemNet, and OrbNet. These models have demonstrated strong generalization in atomistic property prediction and are increasingly adapted for scalar tasks. Including a brief discussion of their relevance—or indicating whether such models are considered for future inclusion—would improve the benchmark’s alignment with current research directions.


2. **Out-of-distribution (OOD) generalization setup**: The current train/test split is well-constructed to avoid data leakage while maintaining distributional similarity. However, it largely reflects interpolation scenarios. To better reflect real-world applications such as materials discovery, where models are often applied to unseen compositional or structural domains, it may be beneficial to additionally provide an OOD-oriented split. For example, holding out particular space groups or element types during training could allow the evaluation of extrapolative generalization capabilities.


3. **Inconsistent fine-tuning strategies across models**: Fine-tuning protocols differ across models—some (e.g., PaiNN, SchNet) use frozen pretrained representations with task-specific heads, while others (e.g., M3GNet, CGCNN) are fine-tuned end-to-end. This heterogeneity complicates interpretation, as performance differences may reflect optimization choices rather than architectural capacity. Clarifying this point further, or optionally including an auxiliary comparison under a shared fine-tuning protocol, would improve the reliability of cross-model comparisons.


4. **Limited input feature diversity**: The dataset targets total ionic conductivity, which in practice can be strongly affected by microstructural factors such as grain boundaries, porosity, and sintering conditions—especially in polycrystalline samples. While such information is understandably difficult to standardize across studies, acknowledging these limitations more explicitly and discussing whether certain processing-related metadata could be incorporated in future versions (even if only for a subset) would help clarify the dataset’s scope and potential extension paths.

**Dataset Code Accessibility:**

Yes

**Dataset Code Comments:**

Yes. Code and data are publicly available with clear structure. All components (CIFs, splits, scripts) are included, and the repository is well-documented.

**Ethical Comments:**

No ethical concerns. The dataset is based on inorganic materials experiments, with no human or sensitive data involved. No foreseeable societal risks.

**Ethical Considerations:**

No, there are no or only very minor ethics concerns

**Final Justification:**

The rebuttal addressed my main concerns regarding dataset limitations, model comparison fairness, and fine-tuning protocols. The planned revisions satisfactorily resolve these issues, and I maintain my positive assessment and rating.

**Limitations Weaknesses:**

While the benchmark is thoughtfully constructed, certain structural limitations in dataset representation and evaluation framing may affect the scope and interpretation of the results.
- The input feature space of the dataset is relatively low in both dimensionality and diversity. Key experimental factors known to strongly influence ionic conductivity in solid-state electrolytes, such as grain boundary density, grain size distribution, porosity, sintering conditions, and other microstructural features, are not included. Although these properties are difficult to standardize or systematically extract from literature, they likely existed in the original experimental context and may have played a critical role in determining the target property. Their absence limits the expressiveness of the dataset and imposes a fundamental ceiling on achievable model performance. Acknowledging this limitation more explicitly, or discussing possible strategies for incorporating such information in the future, would provide a more realistic understanding of the dataset’s utility.
- The performance comparison between geometric GNNs and simpler models such as Random Forest or MLP is structurally imbalanced. While the GNNs are trained on a limited structural subset (with CIF files) and do not utilize explicit global features like composition, space group, or lattice parameters, the conventional models are trained on the full dataset with such features included. This asymmetry in input modality and data scope makes direct performance comparisons potentially misleading. Although the authors acknowledge this issue, the current framing, presenting the models side-by-side without further qualification, may overstate the relative effectiveness of simpler models. A more careful framing or additional controlled comparisons would improve the fairness and interpretability of the benchmark.

**Strengths Contributions:**

Given the scarcity of high-quality experimental datasets in this domain, the contribution is particularly valuable and well-positioned to serve as a standard benchmark for future research.
- The dataset consists of experimentally measured ionic conductivities for lithium-based solid-state electrolytes, paired with structural information. Although not large in size, the data points are valuable given the practical difficulty of obtaining such measurements compared to simulation-based properties.
- The authors took steps to maximize public accessibility of structural data by releasing perturbed CIF files under symmetry constraints, in coordination with ICSD. This reflects a thoughtful and realistic approach to open science.
- Experimental uncertainty is quantified using repeated measurements on nominally identical compositions and structures, offering a rare empirical estimate of the lower bound for predictive error. This grounds model evaluation in realistic conditions, which is often overlooked in benchmark datasets.
- The paper is clearly written and well-structured. The dataset, code, and benchmark setup are openly available and practically usable, supporting reproducibility and broader adoption.

---

> ### Author Rebuttal · Authors · 2025-07-31
>
> Thank you for your very detailed, clear and constructive review. We are happy that you think our contribution is "particularly valuable and well-positioned" to serve as a standard benchmark. Thank you for recognizing the difficulty of assembling an experimental database and for highlighting our efforts in making it as accessible as possible.
>
> ### Comment 1
>
> > The input feature space of the dataset is relatively low in both dimensionality and diversity. Key experimental factors known to strongly influence ionic conductivity in solid-state electrolytes, such as grain boundary density, grain size distribution, porosity, sintering conditions, and other microstructural features, are not included. Although these properties are difficult to standardize or systematically extract from literature, they likely existed in the original experimental context and may have played a critical role in determining the target property. Their absence limits the expressiveness of the dataset and imposes a fundamental ceiling on achievable model performance. Acknowledging this limitation more explicitly, or discussing possible strategies for incorporating such information in the future, would provide a more realistic understanding of the dataset’s utility.
>
> You are absolutely correct and thank you for bringing this up. We will update section 5 with the following paragraph which is largely based on your comment:
>
> >> Varying factors outside the composition and crystal structure including the measurement conditions (frequency, pressure, measuring device, metalization, etc.) and the microstructure (grain size, porosity, phase purity, etc.) that depend on the fabrication process (sintering, cold/hot press, pulverization, heat treatment, etc.) may have important effects on the measured ionic conductivity. The absence of these factors in OBELiX sets a bound to the performance of the models presented here that is partially, but not fully captured by the experimental uncertainty discussed in section 4. The repeated materials presented in Figure 3 could serve as a useful starting point to identify which of these numerous factors have the most impact on the measured conductivity and dictate what additional features could be added to the dataset.
>
> ### Comment 2
>
> > The performance comparison between geometric GNNs and simpler models such as Random Forest or MLP is structurally imbalanced. While the GNNs are trained on a limited structural subset (with CIF files) and do not utilize explicit global features like composition, space group, or lattice parameters, the conventional models are trained on the full dataset with such features included. This asymmetry in input modality and data scope makes direct performance comparisons potentially misleading. Although the authors acknowledge this issue, the current framing, presenting the models side-by-side without further qualification, may overstate the relative effectiveness of simpler models. A more careful framing or additional controlled comparisons would improve the fairness and interpretability of the benchmark.
>
> This is a very good point, models are indeed difficult to compare fairly. The question we used to decide how to present the data was the following: if someone wishes to use *our dataset* to make the best possible prediction of ionic conductivity for a new candidate material, which model should they use? From that perspective the fact that the MLP and the RF *can* take advantage of the full dataset is a strength of these models and the fact that the GNNs don't make explicit use of global features (which can be retrieved from the CIF files) is their own weakness. However your point about overstating the relative effectiveness of simpler models (especially outside the context of OBELiX) is well taken and we will update Figure 4 with the test and validation performance of the MLP and RF when trained only on the CIF subset (which is currently only mentioned in the text). This will make it clear how these models perform given the same conditions as the others.
>
> ### Comment 3
>
> > Missing recent GNN baselines: While the benchmark includes representative models like SchNet, PaiNN, and M3GNet, it omits more recent architectures such as NequIP, MACE, GemNet, and OrbNet. These models have demonstrated strong generalization in atomistic property prediction and are increasingly adapted for scalar tasks. Including a brief discussion of their relevance—or indicating whether such models are considered for future inclusion—would improve the benchmark’s alignment with current research directions.
>
> We chose these model based on their widespread usage, but we agree that there might be newer architectures that may perform better. However, given that the performance gains of these newer model is relatively small compared to models used here (see Ref. 18 for example), we do not expect that they would perform significantly better. For example, PaiNN, which usually outperforms CGCNN and SchNet in these benchmarks, performs equally or worse here.
>
> Moreover, GemNet (2021), OrbNet(2020), MACE(2022) and Nequip(2021) are all more or less contemporary to PaiNN(2022), M3GNet(2022) and SO3Net(2022) based on their Arxiv submission date.
>
> We will add the following discussion to section 4.3 after the 5th paragraph:
>
> >> It is important to note that there may exist more recent GNN architectures that perform better on this task, however given the fact that some of the more recent models tested here still perform equally or close to state-of-the-art models on scalar predictive tasks [18]  we do not believe these newer models would perform significantly better on OBELiX and would likely suffer from the same limitations.
>
> ### Comment 4
>
> > Out-of-distribution (OOD) generalization setup: The current train/test split is well-constructed to avoid data leakage while maintaining distributional similarity. However, it largely reflects interpolation scenarios. To better reflect real-world applications such as materials discovery, where models are often applied to unseen compositional or structural domains, it may be beneficial to additionally provide an OOD-oriented split. For example, holding out particular space groups or element types during training could allow the evaluation of extrapolative generalization capabilities.
>
> This is a really good idea and in the same vein it would be interesting to create a challenge with that set. We do not want to change our current splits at this point because our analysis and benchmark is based on them and, being already public, people may have started training and testing with them, but we are already working on OBELiX* and this could be a great addition to it. We could manually retrieve new entries in domains that we know are underrepresented or not represented at all in OBELiX and since they would be outside existing datasets they would be good candidates for a challenge.
>
> ### Comment 5
>
> > Inconsistent fine-tuning strategies across models: Fine-tuning protocols differ across models—some (e.g., PaiNN, SchNet) use frozen pretrained representations with task-specific heads, while others (e.g., M3GNet, CGCNN) are fine-tuned end-to-end. This heterogeneity complicates interpretation, as performance differences may reflect optimization choices rather than architectural capacity. Clarifying this point further, or optionally including an auxiliary comparison under a shared fine-tuning protocol, would improve the reliability of cross-model comparisons.
>
> We based our fine-tuning strategies on what was recommended in the respective documentations of each model as mentioned at the end of section 4.2.  We agree that this complicates interpretation and that a systematic analysis of tuning strategies would be ideal, so as you suggested we have evaluated the performance of PaiNN and SchNet when fine-tuned end-to-end. We will add the following passage to the paragraph about pretraining is section 4.3:
>
> >> Pretraining of 3D geometric models offers some marginal improvements for PaiNN, SchNet and CGCNN. As mentioned in Section 4.2, the pretraining of PaiNN and SchNet restricts the trainable model size which may reduce accuracy while increasing generalizability. This would explain their slightly higher validation MAE and lower training MAE. **To measure the effect of the fine-tuning strategy alone we also fine-tuned PaiNN and SchNet by allowing all parameters to change. Under this strategy, PaiNN and SchNet had validations MAEs of 1.66$\pm$0.21 eV and 1.81$\pm$0.31 eV while their test MAEs were of 2.60 eV and 2.81 eV respectively. From this limited study, the fine-tuning strategy seems to explain the increased validation MAEs of PaiNN and SchNet in Figure 4. Therefore, in concert with a more restricted fine-tuning strategy,** a better pretrained representation might compensate for the reduced expressivity and increase both accuracy and generalizability, but a much more in-depth analysis of the possible pretraining labels and datasets would be required.

---

> > ### Comment · Reviewer_set9 · 2025-08-03
> > **Reviewer Response to Author Rebuttal**
> >
> > I thank the authors for their detailed rebuttal. The clarifications regarding dataset limitations, model comparison fairness, and fine-tuning protocols satisfactorily address the major concerns I raised, and the planned revisions and supplementary analyses appear appropriate. I appreciate that several of my comments have been directly incorporated into the revised manuscript.
> >
> > Overall, I find the responses reasonable and satisfactory, and I maintain a positive evaluation of the paper.

---

### Official Review · Reviewer_rCUE · 2025-06-22

**Rating:** 2
**Confidence:** 4

**Summary:**

Solid state batteries are in terms of safety and theoretical performance, but the field has struggled to find sufficient materials with high effective ionic-conductivity. In order to avoid using first-principles calculations which are highly inefficient, they introduce a dataset, OBELiX, which contains ionic conductivity and structural data. The hope is that through training a ML model on OBELiX, one could speed up the process of finding effective solid state battery materials.

Existing techniques for finding materials follows a “guess-and-check” approach which is too inefficient considering the first-principles calculations take hours-days per example, and the search-space of potential materials is extremely large.

**Additional Feedback:**

I understand the difficulty of acquiring data from first principles calculations; it is a common issue throughout the field. Nonetheless, the number of new provided samples (246) is too small to be used a dataset publication. I might suggest finding a materials science, nanoengineering, or chemistry venue instead. For a ML dataset, it needs to integrate with existing ML works.

Lastly, it ight be good to mention DFT calculations early in the abstract, introduction, or background. The majority of works that use MLFFs mention it earlier, and as ML researchers  and practitioners in this area, we often use that as a key term to identify relevant work.

**Dataset Code Accessibility:**

Partly

**Dataset Code Comments:**

(1) The provided github code was excellent. The data was clear and understandable, documentation was easily sufficient to understand it. The download is a straight forward as you can get.

(2) The Kaggle was also easy to use, and the download link worked. All required fields were provided.

(3) The code was provided, but saved weights/checkpoints were not. If you specify the seeds used that can also allow for recreation of the model.

**Ethical Considerations:**

No, there are no or only very minor ethics concerns

**Final Justification:**

As the authors and I discussed, my main issue with this dataset is its size. ML for physics/chemical modeling and nano-engineering is my main area of study. I know from experience that 599 incomplete examples is not sufficiently large to provide a substantial increase in quality in a ML model. We look for datasets containing data on the order of 10k as a minimum to fine-tune or train a network. The results the authors show actually support this conclusion: even using their dataset they are not able to beat the established results in literature.

Since the paper focuses almost exclusively on the dataset, I cannot give it a high score. There is no theoretical contribution provided, and the dataset is insufficient for any significant modeling scenario. I understand the difficulty in acquiring data for this field, but that is an established challenge when seeking to publish in this area. Other datasets contain hundreds of thousands or even millions of points.

For this reason, I must keep my rating as a 2.

**Limitations Weaknesses:**

(1) Providing only 599 structures is extremely limited for a machine learning application. While perhaps sufficient for a “classic” ML algorithm, it is not usable to train a neural network. Justification of the benefit of a dataset of this size is required. While it may cover a large portion of the existing data, it is likely to small for use by modern ML techniques.

(1.5) You only introduce 246 new structures out of the 599 total. The retrieval of the other 353 structures from other datasets, while useful, is a much weaker contribution than creating them yourself.

(2) Providing a dataset to encourage new techniques that handle its limitations is suspect reasoning. One should either have a technique they propose or remove the limitation from the dataset.

(3) Your work also contains limited structural information. Only about half of the structures contain it.

(4) Figure 2b only adds up to 597 total entries, yet you claim 599 total structures. Where are the last two?

(5) All the graph NNs you reference are MLFFs. However, you do not say if you are providing trajectories, or just single points.

**Strengths Contributions:**

(1) Justification for this type of work is strong, with MLFFs having shown great value historically.

(2) They provide the first dedicated dataset for this problem at a constant temperature; allowing for easy comparison between candidates.

(3) They provide a recommended training-test split.

(4) Their dataset covers a “large” (no more specificity provided) portion of the current existing results.

(5) The note about uncertainty in Section 4.0 is quite interesting!

---

> ### Author Rebuttal · Authors · 2025-07-31
>
> Thank you for your detailed and constructive feedback. Despite the score, we find your review mostly positive: you mention that our work is strongly justified, that it provides the first easy-to-use dedicated dataset in this field with a relevant train-test split and that some parts of our analysis were very interesting. You rightfully point out that despite its size, our dataset covers most of the data that currently exists. Finally, you took the time to look at our repository (thank you!) and seem to really appreciate our efforts in making it user-friendly.
>
> ### Comment 1
>
> > Providing only 599 structures is extremely limited for a machine learning application. While perhaps sufficient for a “classic” ML algorithm, it is not usable to train a neural network. [...]
>
> We are aware that OBELiX is small and we have been explicit about it in the paper (e.g. in section 5, ``OBELiX is small for ML standards.''). As a reminder, the arguments we make in the paper are:
> 1. It represents a significant portion of all experimental data that exists in this field (as you mentioned in your review)
> 2. It is a common problem in AI for science which highlights the need for frameworks adapted to small multimodal data regimes. (We will come back to this point in our response to your comment 2)
>
> To provide further justification for the usefulness of our datasets for the ML community, we will add the following paragraph to Section 5:
>
> >> [...] Moreover, OBELiX can also be used as a tool to validate and improve molecular dynamics (MD) based methods which are widely applicable across materials science and could later serve as a way to generate a significantly larger computational database of ionic conductivity. For example, in subsequent work, we are currently using a subset of OBELiX to compare the performance of MLFFs and ab-inito methods when predicting ionic conductivity with various MD simulation conditions. Our dataset provides an opportunity to quantitatively test the performance of MLFFs on long timescale MD simulations or ML methods such as LiFlow (arxiv/2410.01464) aimed at accelerating them.
>
> Further, we would like to note that our dataset _is_ usable to train a neural network, as we in fact do by training a multi-layer perceptron (MLP), which achieves the second-best performance of all models.  We are confident that future research, encouraged by the challenges posed by this data set, will provide new neural networks that will at least match the performance of the simpler methods.
>
> ### Comment 1.5
> > You only introduce 246 new structures out of the 599 total. The retrieval of the other 353 structures from other datasets, while useful, is a much weaker contribution than creating them yourself.
>
> We are unsure where you obtained this number. In Figure 2b, there are 246 entries that are part of OBELiX, Laskowski and the ICSD database, is this what you are referring to?
>
> To be clear we did not ``create'' any structure; OBELiX is a dataset of experimentally measured ionic conductivities of real materials found in the literature, structural descriptions are measured experimentally. We gathered incomplete structural information from existing datasets--mostly the Laskowski and LiIon dataset--and manually retrieved the missing information to create a coherent easily accessible dataset. We performed corrections, validation, benchmarking and data analysis on all the entries regardless of how we originally retrieved them.
>
> We believe that gathering and improving data from existing datasets is valuable and it is explicitly mentioned as a suitable contribution in the Datasets & Benchmarks Track scope:
>
> > The scope includes but is not limited to:
> >
> >   - New datasets, **or carefully and thoughtfully designed (collections of) datasets based on previously available data.**
> >   - ...
>
> ### Comment 2
>
> > Providing a dataset to encourage new techniques that handle its limitations is suspect reasoning. One should either have a technique they propose or remove the limitation from the dataset.
>
> Assuming that you are referring to the size of the dataset being the limitation, there are really two things that can be done:
> 1. Increase the size
> 2. Develop methods that work with small data
>
> With regards to 1, as you noted we are providing the first dedicated dataset to this problem, so we are addressing that point by increasing the size from 0 to 599. As you also noted, OBELiX represents a significant portion of existing experimental data, which means that significantly increasing its size would simply require more experimental data.
>
> With regards to point 2, developing algorithms tailored for small data that perform well on this task is a contribution in itself that is separate from creating the dataset. The point of Datasets & Benchmarks Track is to recognize that creating a dataset is a standalone contribution.
>
> ### Comment 3
>
> > Your work also contains limited structural information. Only about half of the structures contain it.
>
> It is true that about half of the entries have crystal description, but it is important to remember that this is the case not because we have not extracted all the information, but because this information is often not measured experimentally. This was not made clear in our paper and the second paragraph of section 3.1 will now read:
>
> >> While the combination of the composition, lattice parameters and space group is often sufficient to qualify materials, they do not fully describe the crystal structure because in general they do not specify the positions of each atom. Some, but not all, experimental papers perform an additional analysis (Rietveld refinement) of the X-ray powder diffraction pattern to estimate atomic positions. Only in these cases is it possible to obtain a full description of the crystal including atomic positions which is necessary to build a crystal information file (CIF). This is why it is not possible to obtain CIF files for all entries in our dataset. The full-crystal description including atomic positions is the information required to perform molecular dynamics simulations or density functional theory calculations, for example.
>
> ### Comment 4
>
> > Figure 2b only adds up to 597 total entries, yet you claim 599 total structures. Where are the last two?
>
> We really appreciate your attention to details on this one. The figure adds up to 597 because there are 2 entries that are not in any of the three datasets on the figure. We will modify the caption of the figure to read:
>
> >> b) Venn diagram showing how OBELiX entries are shared across the ICSD, Laskowski and LiIon datasets. There are 2 OBELiX entries that are not part of any of the three datasets.
>
> ### Comment 5
> > The code was provided, but saved weights/checkpoints were not. If you specify the seeds used that can also allow for recreation of the model.
>
> We have created a pull request on our repository that will add the missing configuration files containing the seeds for MatGl and CGCNN. We also added comments in the RF and MLP scripts explaining how to run each experiment. The seed is specified directly in the scripts. The config files for SchNetPack were already in the main branch. Will merge the pull request at the end of the review process so as not to alter the original state of the repo.
>
> ### Comment 6
>
> > All the graph NNs you reference are MLFFs. However, you do not say if you are providing trajectories, or just single points.
>
> All the GNNs we referenced *can be used as MLFFs* but they are not used as such here. They are used to directly predict ionic conductivity as a property prediction task.
>
> MLFFs *can* be used to make ionic conductivity predictions with molecular dynamics, but these calculations are very involved and resource intensive. Furthermore, the accuracy of such calculations with MLFFs is understudied. We are planning to apply these methods to a small subset of OBELiX in a follow-up publication as mentioned in comment 1
>
> ### Comment 7
>
> > I understand the difficulty of acquiring data from first principles calculations [...] I might suggest finding a materials science, nanoengineering, or chemistry venue instead. [...]
>
> Just to clarify here that OBEliX is not a dataset of first principle calculations, it is a dataset of **experimental** ionic conductivity and structural properties. There are no first principle calculations whatsoever in this study.
>
> As for the venue, we believe that the battery field could benefit from more engagement from the machine learning community and our aim is to make it accessible to them. That is why a good portion of the paper is used to introduce the scientific background. We want to bring the ML community up to speed and introduce a well-posed problem with a dataset to experiment with.
>
> Machine learning is currently used in a broad range of applications, including chemistry, biology, materials science, climate modeling, engineering, astronomy, biodiversity, etc. It is precisely the publication of datasets and methods at venues such as NeurIPS, making the specifics of each domain accessible to ML researchers, that has driven the success of ML in these areas, as well as encouraged new ML research directions.
>
> ### Comment 8
>
> > Lastly, it might be good to mention DFT calculations early in the abstract, introduction, or background. [...]
>
> As mentioned in response to your previous comment, there is no DFT involved in this dataset. However, OBELiX is indeed a great tool to validate methods such as ab initio molecular dynamics (AIMD) which do use density functional theory (DFT). We mention AIMD in the introduction, but in view of your comment we will explicitly mention that it is based on DFT:
>
> >> [...] Computational discovery, on the other hand, requires time-consuming atomistic simulations, such as ab initio molecular dynamics (AIMD) which is based on density functional theory (DFT), to accurately capture the complex relationship between ionic conductivity and the material's structure and composition. [...]

---

> > ### Comment · Reviewer_rCUE · 2025-08-01
> > **Response to Rebuttal**
> >
> > Key points:
> > 1 - I understand the creation of a dataset is a standalone contribution, but this dataset cannot be used feasibly in machine learning. It is the best dataset for this specific task, but that does not mean it is a useful dataset for machine learning.
> >
> > 2 - I think that this dataset is useful in nanoengineering, physics, and chemistry domains. This however is a ML/DL venue; all publications must be focused towards usefulness in ML.
> >
> > Comment 1:
> > I appreciate the authors’ recognition that this dataset does not match with current ML/DL standards for size. I understand that it is the largest dataset available, and that acquiring data is extremely difficult. However, the fact remains that it is too small to be useful in ML applications. Other datasets for related problems are several orders of magnitude larger than OBELiX, and are still considered to be very small by DL standards.
> >
> > The justification by fine-tuning with an MLP with this dataset is insufficient. This model still did not achieve state of the art performance, and is not shown to be useful in practice. Perhaps training with a newer foundation model like MatterSim could provide better results, and then a possible justification could be the ability of the model to generate correct results that could not be made by any existing model without the use of this data. As a side note, in reduced data settings, it is often better to fine-tune a subset of parameters rather than the whole model.
> >
> > The need for techniques that handle small datasets has been well established for quite some time… providing more examples of situations that can benefit from a non-existing technique is not a substantial enough contribution to warrant a publication.
> >
> > Comment 1.5:
> > Apologies for the confusion; I appreciate the clarification. This is actually a weakening argument for your paper. While retrieving data from other sources and compiling it to a single location is useful, it is a bit weaker than generating new data yourself.
> >
> > Comment 2:
> > This returns the core issue: just because the dataset size is increased does not mean it is a substantial enough increase to warrant a paper. Otherwise, every new experimental structure could be used as a new publication. The dataset would increase from 599 to 600, then 600 to 601…
> >
> > Having dealt with it myself, I understand how difficult it is to acquire data. Perhaps using simulated trajectories in addition to experimental trajectories could ease this burden.
> >
> > I understand the creation of a dataset is a standalone contribution, but it needs to be a dataset that actually can be used in machine learning.
> >
> > Comment 3:
> > This is an understandable limitation, but is a limitation nonetheless.
> >
> > Comment 4, 5, 6, 8:
> > Thanks for adding the clarity! I appreciate the extra information,
> >
> > Comment 7:
> > I agree that ML for nanoengineering is an underdeveloped field, however existing ML researchers will not be inspired by a dataset that is too small. I asked others if they would consider training./fine-tuning a machine learning model with a dataset less than a 1000 points and they unanimously responded that they would not.

---

> > > ### Author Response · Authors · 2025-08-04
> > >
> > > Thank you for your response and for engaging in the conversation.
> > >
> > > We think it is fair to say that your main claim is that:
> > >
> > > > The dataset is not useful for machine learning because it is too small.
> > >
> > > "Usefulness" is subjective and dependent on the perspective. We are part of the machine learning community and we (obviously) find it useful. We are currently using it to train and develop ML models and we are using it (as mentioned before) to evaluate the performance of different machine learning force fields when running molecular dynamics.
> > >
> > > What is large enough to be ML-worthy is also somewhat arbitrary. For example, what size would you consider large enough? There is no mention of minimal size in the call for papers and there have been datasets of various sizes accepted in the past 4 editions.
> > >
> > > Even If you don't believe one can ever train successfully on OBELiX because of its size, do you disagree that it is _useful_ as a benchmark? There are numerous examples of LLM benchmarks that are several orders of magnitude smaller than what LLMs are trained on and they are still useful to evaluate their performance and improve them.
> > >
> > > Finally, you seem to think that our paper and datasets are of good quality and they are useful in general. You have also acknowledged our efforts in addressing your comments. It seems like your rejection is based mainly on your perception of usefulness for the ML community in general. In that context, we ask you to consider if your current score reflects that fairly: 2, "a paper with technical flaws, weak evaluation, inadequate reproducibility and incompletely addressed ethical considerations"?

---

> > > > ### Comment · Reviewer_rCUE · 2025-08-04
> > > > **Continuation of Discussion**
> > > >
> > > > Yes, my main issue is that the dataset is insufficient for use in ML applications. You are indeed correct that usefulness is subjective, that is why we have reviewers in the first place! It is our job to determine how beneficial a publication is to the community. Indeed, a “dataset” with a single example would not be considered viable, but a dataset with 100k examples would be considered a strong publication.
> > > >
> > > > I am not arguing that there is no use to OBELiX; indeed I have frequently acknowledged the difficulty and acquiring physics datasets, as well as the work that goes into curating datasets. This publication is simply not setup for a ML dataset, it is structured more as a physical-science dataset. I encourage the authors to continue with this work, and perhaps to seek publication in a physical science venue. This however is a machine learning publication; submissions need to be made to fit ML, not the other way around.
> > > >
> > > > As a last note, you mentioned that many of the examples are incomplete, meaning the effective dataset size for any given problem is smaller than the actual 599 examples.

---

> > > > > ### Author Response · Authors · 2025-08-05
> > > > >
> > > > > We have prepared this publication entirely for an ML venue and audience from the start: We have introduced every relevant material science concept, we provided splits and data analysis that is tailored for ML. We have benchmarked several ML models with different pre-training methods and modified architectures. We have followed the best practices in the field to distribute the dataset. You have recognized and even praised some these efforts. In that context, we think it is a bit unfair to say that "[t]his publication is simply not setup for a ML dataset". We did not merely submit a physics paper to this venue hoping that it would fit.
> > > > >
> > > > > The chief value of this (and other data sets) is that we can use it to train ML models and then make predictions about new data. If you were to use machine learning to work on the discovery of new solid-electrolytes, perhaps you would consider using Bayesian optimisation, active learning and/or generative models to learn to propose new candidates. If that was the case, would you prefer to start from no data whatsoever, or from an organised data set containing most of the existing materials that have been experimentally annotated, regardless of the size.
> > > > >
> > > > > > [A] “dataset” with a single example would not be considered viable, but a dataset with 100k examples would be considered a strong publication.
> > > > >
> > > > > Between 1 and 100k, where is the threshold of amount of examples for a "strong publication"? Is it really the number of examples that makes it a strong publication? Isn't the number of examples task-dependent?
> > > > >
> > > > > We strongly believe that OBELiX has its place in the Dataset and Benchmark Track, but you remain convinced that its size makes it unsuitable for an ML venue. This is a fundamental point that we cannot change at this point. We hope that you can revise your score to reflect your opinion of the overall quality of the paper, not just the size of the dataset, but we understand if you don't. Thank you for the respectful conversation.

---

### Official Review · Reviewer_63rw · 2025-07-04

**Rating:** 4
**Confidence:** 3

**Summary:**

This paper introduces OBELiX, a curated dataset of 599 solid-state electrolyte materials with experimentally measured room-temperature ionic conductivities, structural descriptors (space groups, lattice parameters), and full crystallographic information (CIF files) for 321 structures. The dataset addresses the scarcity of experimental ionic conductivity data for machine learning (ML) applications in solid-state battery research. The authors benchmark seven ML models—including graph neural networks (GNNs) and traditional methods—finding that simple models like Random Forest (RF) and Multilayer Perceptron (MLP) outperform GNNs, likely due to dataset size constraints and challenges in handling partial atomic occupancies. The work emphasizes the need for ML approaches tailored to low-data regimes and provides carefully designed training/test splits to avoid data leakage.

**Dataset Code Accessibility:**

Yes

**Dataset Code Comments:**

The dataset can be accessed

**Ethical Considerations:**

No, there are no or only very minor ethics concerns

**Final Justification:**

This paper provides a well-curated and valuable dataset for machine learning research in solid-state electrolytes, with careful benchmarking and data splitting to ensure robust evaluation; however, the dataset is still relatively small, which limits the performance and generalizability of complex models. Therefore, I recommend borderline accept.

**Limitations Weaknesses:**

1. With 599 entries, OBELiX remains small by ML standards, potentially constraining the performance of complex models and hindering generalizability to broader material spaces.
2. Only 54% of entries include CIF files, and atomic positions for some structures are obscured by noise (to comply with ICSD restrictions), which may limit structural feature extraction.
3. Large standard deviations in cross-validation MAE (e.g., up to 0.68 for pretrained SchNet) indicate high sensitivity to hyperparameters and data splits, challenging reliable model comparison.

**Strengths Contributions:**

1. OBELiX integrates data from existing sources (e.g., Liverpool Ionics, Laskowski) with manual curation, ensuring consistency in structural and conductivity measurements. The inclusion of CIF files and partial occupancy information enhances its utility for detailed material modeling.
2. The study systematically evaluates diverse ML models using 5-fold cross-validation, highlighting performance gaps between simple and complex architectures. This provides clear baselines for future research.
3. The dataset’s training/test splits are designed to avoid leakage by grouping entries from the same paper or with identical compositions, ensuring robust model evaluation.

---

> ### Author Rebuttal · Authors · 2025-07-31
>
> Thank you for your clear and straightforward review. We are happy that you recognized the value of our contribution in curating the OBELiX dataset, that you think our evaluations "provides clear baselines for future research" and that you think our test split ensures "robust evaluation".
>
> ### Comment 1
>
> > With 599 entries, OBELiX remains small by ML standards, potentially constraining the performance of complex models and hindering generalizability to broader material spaces.
>
> As you know, we are aware of this fact and have been explicit about it in the paper (e.g. in section 5, ``OBELiX is small for ML standards.''). As a reminder, the arguments we make in the paper are that:
> 1. It represents a significant portion of all experimental data that exist in this field
> 2. It is a common problem in AI for science which highlights the need for frameworks adapted to small multimodal data regimes (see Xu et al. npj Computational Materials 9, 42 (2023) for example).
>
> Now, with respect to your point about complex models and applicability to broader materials spaces, we will add the following sentences to the second paragraph of section 5:
>
> >> [...] Moreover, OBELiX can also be used as a tool to validate and improve molecular dynamics (MD) based methods which are widely applicable across materials science and could later serve as a way to generate a significantly larger computational database of ionic conductivity. For example, in subsequent work, we are currently using a subset of OBELiX to compare the performance of MLFFs and ab-initio methods when predicting ionic conductivity with various MD simulation conditions. Our dataset provides an opportunity to quantitatively test the performance of MLFFs on long timescale MD simulations or ML methods such as LiFlow (arxiv/2410.01464) aimed at accelerating them.
>
>
> ### Comment 2
>
> > Only 54\% of entries include CIF files, and atomic positions for some structures are obscured by noise (to comply with ICSD restrictions), which may limit structural feature extraction.
>
> It is true that not all structures have a full-crystal description, but it is important to remember that this is the case not because we have not extracted all the information, but because this information is often not measured experimentally. This was not made clear in our paper and the second paragraph of section 3.1 will now read:
>
> >> While the combination of the composition, lattice parameters and space group is often sufficient to qualify materials, they do not fully describe the crystal structure because in general they do not specify the positions of each atom. Some, but not all, experimental papers perform an additional analysis (Rietveld refinement) of the X-ray powder diffraction pattern to estimate atomic positions. Only in these cases is it possible to obtain a full description of the crystal including atomic positions which is necessary to build a crystal information file (CIF). This is why it is not possible to obtain CIF files for all entries in our dataset. The full-crystal description including atomic positions is the information required to perform molecular dynamics simulations or density functional theory calculations, for example.
>
> Furthermore, we would like to note, as is done in the paper, that the fact that different sources of data contain different portions of each data point should also encourage machine learning research tailored to tackle this situation. For example, typical GNNs are not able to handle incomplete data.
>
> As for the noise, we have shown in Table S3 that it has little to no effect on model performance. This is probably because keeping the symmetry intact imposes a strong restriction on how the noise-which is already small-affects the material.
>
> ### Comment 3
>
> > Large standard deviations in cross-validation MAE (e.g., up to 0.68 for pretrained SchNet) indicate high sensitivity to hyperparameters and data splits, challenging reliable model comparison.
>
> It is true that the standard deviation of the MAE across the 5-folds is relatively high for some of the models. We believe that this is a result of (1) the size of the dataset and (2) the fact that validation folds are chosen randomly within the training set. This creates a situation where some of the folds may end up with a set of materials that is significantly different from the others (similarly to how we built our test set) which increases the validation MAE of that fold.
>
> We chose random validation splits because our goal was to reproduce a typical training workflow and to show the importance of carefully choosing test-train splits in this case (we discuss this in the 5th paragraph of section 4.3). We completely agree that it indicates high sensitivity to hyperparameters including how one chooses the splits. To clarify this point we added the following paragraph at the end of section 4.3:
>
> >> It is important to bear in mind that the variability of the prediction accuracy is high in this small data regime as illustrated by the validation MAEs’ standard deviations and that much of the difference between models falls within that variability. Performance is dependent on the (random) choice of cross-validation splits which ultimately dictate the choice of hyperparameters. Complex GNNs with more hyperparameters are more prone to overfitting hyperparameters to a specific set of splits which makes them particularly difficult to tune and compare. Indeed, the variability across folds is smaller for the RF and MLP than for the GNNs.
>
> We believe that this is part of what one can conclude from our results because of the way we designed our experiments. In other words, it is a limitation of having to do statistics on fewer data and a limitation of current geometric GNNs with many hyperparameters when applied to this regime, but we don't see it as a limitation of our study per se which is why we did not include it in section 5.

---

> > ### Comment · Reviewer_63rw · 2025-08-06
> > **Response to the rebuttal**
> >
> > The author has resolved the issue I raised, but there is indeed a problem with the dataset not being large enough. Nevertheless, it is still a good contribution to the field, so I will keep score.

---

### Decision · Program_Chairs · 2025-09-18

**Decision:**

Reject

**Comment:**

This paper has been intensively discussed. The main issue is a reviewer is concerned that the dataset is not generic and thus not very useful for AI/ML. I feel this is a valid concern and thus lean towards reject.